Correlations between personality traits and specific groups of alpha waves in the human EEG

Johannisson Tomas tomas.johannisson@outlook.com
Department of Psychiatry, Sahlgrenska University Hospital , Mölndal , Sweden
Yuan Tifei
Electronic publication date: 2016 Jul 19
Publication date: 2016
Volume: 4
Electronic Location ID: e2245
Received 2016 Mar 16; Accepted 2016 Jun 22
Copyright: ©2016 Johannisson
Copyright year: 2016
Copyright holder: Johannisson
License: This is an open access article distributed under the terms of the Creative Commons Attribution License, which permits unrestricted use, distribution, reproduction and adaptation in any medium and for any purpose provided that it is properly attributed. For attribution, the original author(s), title, publication source (PeerJ) and either DOI or URL of the article must be cited.
License URL: https://creativecommons.org/licenses/by/4.0/

Keywords: Extraversion, Neuroticism, Alpha waves, Temperament, IPIP-NEO, Cerebral cortex, Columnar organization, Alpha hypothesis

Funding: The author received no external funding for this work.

==============================
Background. Different individuals have alpha waves with different wavelengths. The distribution of the wavelengths is assumed to be bell-shaped and smooth. Although this view is generally accepted, it is still just an assumption and has never been critically tested. When exploring the relationship between alpha waves and personality traits, it makes a huge difference if the distribution of the alpha waves is smooth or if specific groups of alpha waves can be demonstrated. Previous studies have not considered the possibility that specific groups of alpha waves may exist.

Methods. Computerized EEGs have become standard, but wavelength measurements are problematic when based on averaging procedures using the Fourier transformation because such procedures cause a large systematic error. If the actual wavelength is of interest, it is necessary to go back to basic physiology and use raw EEG signals. In the present study, measurements were made directly from sequences of alpha waves where every wave could be identified. Personality dimensions were measured using an inventory derived from the International Personality Item Pool.

Results. Recordings from 200 healthy individuals revealed that there are three main groups of alpha waves. These groups had frequencies around 8, 10, and 12 waves per second. The middle group had a bimodal distribution, and a subdivision gave a total of four alpha groups. In the center of each group, the degree of extraversion was high and the degree of neuroticism was low. Many small differences in personality traits were found when the centers were compared with one another. This gave four personality profiles that resemble the four classical temperaments. When people in the surrounding zones were compared with those in the centers, relatively large differences in personality traits were found.

Conclusions. Specific groups of alpha waves exist, and these groups have to be taken into account when correlations are made to personality dimensions and temperament types. There is a link between alpha waves and personality traits, and this link implies that there is an underlying relationship. To explain the nature of this relationship, there are two hypotheses that can be applied. One of these deals with the general organization of the forebrain and the other explains why the brain generates alpha waves.

Introduction

Alpha waves and personality dimensions

Many researchers have attempted to define the relationship between alpha waves and personality dimensions (Gale, 1983; Klimesch, 1999). Of all the data obtained via electroencephalography (EEG), individual alpha frequency stands out as a measure that can be correlated with personality dimensions. For instance, Robinson (2001) found 11 studies that reported a link between alpha frequency and the degree of extraversion. In all of these studies, the mean value of the alpha frequency was slightly higher in groups with introverted individuals compared to groups with extraverted individuals. The differences were statistically significant in seven of the studies.

Over the years, controversies have emerged regarding the various factors that could influence the link between alpha waves and extraversion (Gale, 1983; Tran, Craig & McIssac, 2001; Robinson, 2001). In a study conducted by Hagemann et al. (2009), internal and external factors were found to have a minimal effect on this correlation. The authors tested whether there was a linear relationship between alpha power density and extraversion, and they found that the correlation coefficient for the overall measure was small (0.22). None of the previous studies tested for a relationship along curved lines or took into consideration that there might be specific alpha groups (Gale, 1983; Klimesch, 1999; Tran, Craig & McIssac, 2001; Robinson, 2001; Hagemann et al., 2009).

Need for an alpha hypothesis

To understand how alpha waves are linked to personality traits, it would be helpful to have an explanation as to why the brain generates alpha waves. A possible explanation involves the activity in the columns in the cerebral cortex. It has been suggested that each column exhibits neuronal activity either at a high level or at a comparatively low level (Johannisson, 1984).

In this model, a cerebral column has activity at the high level for a short period of time. When the activity in a column changes from the high level to the low level, the activity in another column may jump up to the high level. Thus, there is a continuous turnover of highly active columns, which results in a flow of thoughts and feelings.

There is a difference in mental activity when something is appearing in consciousness and when that something has disappeared from the conscious level. Two different levels of neuronal activity may account for this observation. Furthermore, only a very tiny fraction of everything that could appear in consciousness actually appears there at each moment in time. To account for this, it is likely that only a very small number of columns are involved in high-level processing at any one time (Johannisson, 1993).

In the “two levels” hypothesis, the number of columns active at the high level is kept within certain limits via a regulating system (Johannisson, 1984). When the number of highly active columns is at the lower limit, regulating signals from the thalamus raise the excitability of all columns that are active at the low level until one of them jumps up to the high level of activity. Conversely, when the upper limit is reached, the regulating system decreases the excitability. When one of the highly active columns leaves the high conscious level, the system returns to the lower limit, such that the excitability will be increased again. These repeated changes in excitability in all columns with low activity are seen in the EEG as alpha waves. This hypothesis explains why and how alpha waves are generated (Johannisson & Nilsson, 1996).

Groups of alpha waves

The limits for the number of highly active columns can be different for different individuals (Johannisson & Nilsson, 1996). Therefore, if the hypothesis is true, there must be specific frequency groups corresponding to the different limits. However, alpha frequencies from different individuals are usually described as having a distribution that is smooth and homogeneous (Klimesch, 1999; Robinson, 2006; Başar, 2012; Bazanova & Vernon, 2014). This is an assumption that has become generally accepted, although it has never been critically tested.

In the present study, the distribution was not at all smooth, and alpha waves from different individuals were found to be in three large groups. The alpha frequencies in these groups were around 8, 10, and 12 Hz (waves per second). The groups are very easy to miss. In a histogram showing the distribution of individual alpha frequencies, the groups are visible only if the bin width is narrow.

The existence of three alpha groups was predicted from an alpha hypothesis (Johannisson & Nilsson, 1996), and the present study was designed in such a way that it was possible to test if specific alpha groups exist. The experiment is very simple but somewhat time-consuming because a narrow bin width in a histogram requires a relatively large number of observations. When the alpha frequencies from 200 individuals were included in the histogram, the pattern with three large alpha groups became clearly visible.

The link between alpha waves and personality dimensions was explored further in the present study. The new finding of distinct alpha groups has to be taken into account when studying how the alpha waves are related to personality dimensions and temperament types.

Methods

Participants

There were 200 participants in the study, and to reach this number, 204 healthy individuals were invited to take part. Approval was obtained from the Regional Ethics Board for University of Göteborg (R627–97) and written informed consent was obtained from all participants. For participants younger than 18 years, written informed assent was also obtained from a parent. The age of the participants ranged from 15 to 70 years, with a mean age of 37.5 years and a standard deviation of 13.0 years. The study included 63% females and 37% males. Of the participants, 34% were students, 26% nurses, 14% doctors, 7% social workers, and 19% were individuals with other occupations.

Figure 1 Alpha waves.

(A) Short sequences of alpha waves. (B) An example of a sequence used in the present study. Blue tracings are from AFz–TP9 and red tracings are from AFz–TP10.

EEG measurements

The individual alpha frequency was measured from single sequences of alpha waves where every wave could be identified (Fig. 1). When testing the alpha hypothesis (Johannisson & Nilsson, 1996), it is the actual wavelength that is of interest, rather than the amount of oscillatory components in different frequency bands. For the purpose of this study, measurements directly from sequences of alpha waves have a much higher accuracy compared to data from averaging procedures using the Fourier transformation. A comparison between the two methods is made in ‘Discussion’.

Figure 1 shows how the wavelength and alpha frequency were calculated. The number of waves in a sequence of alpha waves divided by the time for the whole sequence gave the alpha frequency. One sequence of alpha waves was used for each participant, and to ensure a reasonably high degree of objectivity, the longest sequence was selected. For most of the recordings, this procedure left very little room for subjective judgments. Sequences of alpha waves from all participants are available in Data S1.

Spontaneous activity was recorded while the participants rested in a reclining chair with their eyes closed. Recording took place in a silent room with reduced lighting. The recording conditions were optimized to enable the collection of clear and clean alpha waves. The electrodes were placed on hairless areas of skin behind the ears and on the midline just below the hairline. This made it possible to use tape electrodes that resulted in a considerably lower degree of discomfort compared to ordinary arrangements. Moreover, the probability of getting long sequences of alpha waves was expected to increase when the participant could relax during the recording session. This also ensured that muscle activity seldom interfered with the recording. A small amount of conductive gel was used and the impedance values were below 3 kOhm for most participants.

Data were generally collected during a total time of 2.5 min (30 epochs of 5 s each, as shown in Fig. 1). When there was a large amount of irregular activity, the recording time was prolonged to increase the chance of obtaining a long sequence of alpha waves. Two channels were used, corresponding to AFz-TP9 and AFz-TP10.

In a pilot study with EEGs from 213 subjects, different recording sites were compared in a methodical way (T Johannisson, 1999, unpublished data). A few recording electrodes were found to be sufficient to measure the alpha frequency because alpha waves are widespread. For instance, the longest sequence of regular waves was found in the occipital region in approximately half of the recordings, and in 96% of these recordings, the same alpha waves were also seen in other regions. When the longest sequence appeared in a non-occipital region, 88% of the EEGs had the same alpha waves recorded simultaneously from occipital electrodes.

In histograms showing the distribution of individual alpha frequencies, distinct groups are visible only if the bin width is narrow. This requires a fairly high number of participants. In the pilot study, various combinations of bin width and number of individuals were tested. A clear pattern in the histogram required recordings from at least 200 subjects.

Personality measurements

The IPIP-NEO inventory was derived from the International Personality Item Pool, 2016 (http://www.ipip.ori.org). The 120-item version of the IPIP-NEO (Johnson, 2014) was used, and all participants completed all items. An online analysis gave estimates with respect to five broad domains and 30 subdomains of personality (http://www.personal.psu.edu/j5j/IPIP; Johnson, 2016). After adjusting for age and gender, the scores from the questionnaire were converted into percentile estimates.

Compared with the reference data used in the online analysis, the participants in the present study had, on average, higher extraversion scores and lower neuroticism scores. Moreover, many of the participants in the present study had high scores in the trust and cooperation subdomains.

The differences described above may reflect an unintentional bias in the recruitment of participants for this study. To facilitate comparison with future studies, the estimates are not normalized to fit the total mean values in the present sample. The data from the personality measurements are provided in Data S2.

Results

The degree of extraversion and neuroticism is plotted against the individual alpha frequency in Fig. 2. The correlation lines in this figure are the usual way of describing the relationship between personality traits and alpha waves. However, there is another better way if we take into consideration that specific groups of alpha waves may exist.

Figure 2 Scatter diagrams showing alpha frequency versus extraversion and neuroticism.

Data for all 200 participants in the present study. The correlation coefficient for extraversion was −0.16 (statistically significant, p < 0.05 two tails) and the correlation coefficient for neuroticism was 0.02 (not statistically significant).

Three main alpha groups

The histogram in Fig. 3A shows the distribution of alpha frequencies from different individuals. The distribution of the data is not smooth and large groups are visible. The center of one group is slightly above 8 Hz and the center of another group is somewhat below 12 Hz.

Figure 3 Distribution of individual alpha frequencies.

The histograms have a bin width of 0.1 Hz. (A) All 200 participants. The recordings include poorly, intermediately, and well developed alpha waves. (B) As a control, recordings with poorly developed alpha waves are excluded. This reduced the sample size to n = 150. (C) Only recordings with well developed alpha waves (n = 69).

The data ranging from 9 to 11 Hz can be described as one large group with a bimodal distribution. A subdivision of the main group around 10 Hz gives a total of four groups, where the subgroups have alpha frequencies around 9.5 and 10.5 Hz.

Many participants did not have long sequences of well developed alpha waves (details can be found in Data S1). To test whether the existence of the alpha groups depends on how well the alpha waves are developed, the recordings were classified into three types: poorly, intermediately, and well developed alpha waves.

When the alpha waves are poorly developed, measurements of the individual alpha frequency become problematic. All such measurements are excluded from Fig. 3B, and yet the three main alpha groups and the bimodal distribution for the middle group remain.

When the participants with an intermediate degree of alpha waves are also excluded, leaving only recordings with well developed alpha waves, as shown in Fig. 3C, the group around 12 Hz becomes small and the bimodal distribution for the middle group is not visible. What remains are a distinct group at 8 Hz and a gap at 9 Hz. Relatively few participants are included in Fig. 3C, and this histogram illustrates the problem with a narrow bin width and a small number of subjects.

Three alpha segments

In Fig. 4, the three main alpha groups are studied one at a time. Furthermore, mean values for the personality estimates are used, with these mean values reducing the number of data points. For example, there are four data points at 7.9 Hz in Fig. 2A and the mean value of the estimates from these four participants is shown as one data point in Fig. 4A.

Figure 4 Three alpha segments.

The frequency range was divided into three segments and the personality estimates are shown as mean values. (A–C) Extraversion. (D–F) Neuroticism. The trend lines are second-order polynomials.

The relationship between the alpha frequency and personality dimensions is much better described by curved lines than by straight lines (Fig. 4). The group around 8 Hz has a high degree of extraversion at the center of the group but not in the surrounding zones (Fig. 4A). This pattern is also seen for the groups around 10 and 12 Hz (Figs. 4B–4C).

The curves for neuroticism (Figs. 4D–4F) indicate that there is a difference between the centers of the groups and the surrounding zones. Overall, Fig. 4 supports the existence of three main alpha groups and substantiates the notion that extraversion is high and neuroticism is low in the center of each group.

Broad domains

Center versus surrounding area

To test whether there was a statistically significant difference between the center and the surrounding area, the data from the surrounding zones in Fig. 4 were combined. The selected ranges were 7.05–7.65, 8.55–9.35, 10.65–11.35, and 12.15–12.55 Hz, and the sample size for the surrounding data was n = 43. The ranges for the center data were 7.85–8.15, 9.85–10.15, and 11.65–11.95 Hz, which resulted in a sample size of n = 49. Two-tailed t-tests were used to determine differences.

For extraversion, the difference between the center and the surrounding area was highly significant (p < 0.001). The difference was also statistically significant for neuroticism (p < 0.01) and agreeableness (p < 0.05), but not for conscientiousness. For openness to experience, the mean estimates were about the same for the center and the surrounding area (Fig. 5A).

Figure 5 Mean estimates of personality traits in five domains.

(A) Center and surrounding area. (B) Three alpha groups. (C) Four alpha groups.

Three alpha groups

Figure 5B shows a comparison among the mean estimates for three groups of alpha waves. To ensure an adequate sample size when the three groups were considered separately, wide ranges for the center data were used (7.75–8.45, 9.45–10.55, and 11.45–12.05 Hz). The sample sizes of these three groups were 34, 68, and 28.

The differences among the three groups in Fig. 5B were relatively small, but there were some interesting trends. The group at 12 Hz had the highest score for all domains except for extraversion. For openness to experience, the difference between the groups at 8 and 12 Hz was statistically significant (p < 0.05).

Four alpha groups

The four groups shown in Fig. 5C had ranges of 7.95–8.35, 9.35–9.85, 10.15–10.65, and 11.45–12.05 Hz. These limits gave sample sizes of 26, 33, 34, and 28. The ranges for the four groups were sufficiently narrow when searching for personality traits that are representative of the centers of the alpha groups.

Figure 5C shows how the data were transformed after the division of the main group around 10 Hz into subgroups around 9.5 and 10.5 Hz. For neuroticism, the group at 9.5 Hz had the lowest score and the group at 10.5 Hz had the highest score of the four groups. For agreeableness, the group around 10.5 Hz stood out among the other groups.

The group at 9.5 Hz had the highest score on conscientiousness, but the score was just slightly higher than that obtained from the group at 12 Hz. The latter group still had the highest score on openness to experience and the lowest score on extraversion.

Subdomains

Each of the five broad domains has six subdomains (Table 1). Most of the differences among the four alpha groups were small, but when taken together they corresponded to four different personality profiles.

Table 1 Thirty subdomains from the five-factor model (Johnson, 2014).

The four alpha groups (8, 9.5, 10.5, and 12) were defined by the alpha frequency in the same way as in Fig. 5C.

		8	9.5	10.5	12	
Extraversion	Friendliness	79	70	66	65	
	Gregariousness	79	68	69	76	
	Assertiveness	43	49	49	42	
	Activity level	38	48	45	47	
	Excitement-seeking	39	40	37	34	
	Cheerfulness	66	62	60	54	
Neuroticism	Anxiety	30	23	35	22	
	Anger	37	25	32	31	
	Depression	29	29	35	39	
	Self-consciousness	25	37	32	34	
	Immoderation	34	36	40	36	
	Vulnerability	37	33	39	38	
Agreeableness	Trust	73	74	68	74	
	Morality	64	67	64	61	
	Altruism	54	61	56	64	
	Cooperation	70	77	66	78	
	Modesty	63	52	55	61	
	Sympathy	72	73	68	70	
Conscientiousness	Self-efficacy	38	51	50	47	
	Orderliness	55	59	58	59	
	Dutifulness	52	67	61	59	
	Achievement-striving	47	45	47	48	
	Self-discipline	60	59	57	59	
	Cautiousness	48	63	55	63	
Openness to experience	Imagination	16	24	26	29	
	Artistic interests	39	47	43	53	
	Emotionality	56	53	56	63	
	Adventurousness	70	61	60	63	
	Intellect	45	57	54	59	
	Liberalism	65	72	69	70	

8 waves per second — Friendly and impulsive

The group around 8 Hz had a very high score in the friendliness subdomain. This score was significantly higher compared to the score for the group at 12 Hz (p < 0.02). The group at 8 Hz also had a mean score that was clearly higher than that of the group at 10.5 Hz (p < 0.02). The difference was smaller when the group at 8 Hz was compared with the group at 9.5 Hz, although it was still statistically significant (p < 0.05).

There were several other subdomains that could be used to characterize the group at 8 Hz. In the cautiousness subdomain, the group around 8 Hz had a lower mean score than did the groups around 9.5 Hz (p < 0.05) and around 12 Hz (p < 0.05). The difference between the group at 8 Hz and the group at 10.5 Hz was not large enough to be statistically significant.

9½ waves per second — Peaceful and reliable

In the anger subdomain, the group around 9.5 Hz had the lowest score of the four groups. The difference was largest when compared with the group at 8 Hz, although this result did not reach statistical significance.

The group at 9.5 Hz had relatively high estimates in almost all of the conscientiousness subdomains. In the dutifulness subdomain, this group had a higher score than the group around 8 Hz, and this difference was statistically significant (p < 0.05). Also, when compared with the groups at 10.5 and 12 Hz, the group at 9.5 Hz had a higher mean score, although these differences were not statistically significant.

10½ waves per second — Worried and less agreeable

People in the group at 10.5 Hz had a mean score in the anxiety subdomain that was higher than that in the groups at 9.5 Hz (p < 0.05) and at 12 Hz (p < 0.05). When compared with the group at 8 Hz, the group at 10.5 Hz had a somewhat higher score, but the difference was not statistically significant.

In the agreeableness subdomains, while the group at 10.5 Hz did not have low scores, the other three groups generally exhibited scores that were slightly higher. The difference between the group around 10.5 Hz and the group around 12 Hz in the cooperation subdomain was statistically significant (p < 0.05).

12 waves per second — Less happy and more open

The group at 12 Hz had a somewhat lower score on cheerfulness and a slightly higher score in the depression subdomain than the other groups, but these differences were not large enough to be statistically significant. There was a tendency for the group at 12 Hz to have a more open cognitive style compared to the other three groups. The subdomains associated with openness to experience in Table 1 provide more detailed information than does the broad domain shown in Fig. 5C.

Happiness

The cheerfulness, anxiety, and depression subdomains are all related to happiness. The three curves in Fig. 6A indicate that a high degree of happiness can be found in the center of the group around 8 Hz.

Figure 6 Curved lines representing estimates for the cheerfulness, anxiety, and depression subdomains.

The curves, which are second-order polynomials, were generated via polynomial regression. Light-colored rectangles denote four groups of alpha waves.

The main group around 10 Hz had a reasonably high degree of happiness in a frequency range that was much wider than the ranges for the groups around 8 and 12 Hz (Fig. 6). Thus, there was sufficient space for the subgroups at 9.5 and 10.5 Hz.

People in the main group at 10 Hz had a curve for depression that was shifted slightly to the right when compared with the curve for anxiety (Fig. 6B). This was also seen for people in the group at 8 Hz (Fig. 6A).

In the cheerfulness subdomain, the maximum level for the group at 12 Hz was somewhat lower compared to that for the other groups (Fig. 6). The line for depression in the group at 12 Hz did not have a bend in the same direction as the other groups (Fig. 6). When compared with those in the center of the group at 12 Hz, people in the surrounding zones had much higher anxiety scores (Fig. 6C).

Discussion

The primary aim of the present study was to test the two levels hypothesis (Johannisson, 1984). This hypothesis deals with the general organization of the forebrain, and, if it withstands repeated testing, it may contribute to our understanding of the relationship between mental activity and brain activity.

The two levels hypothesis

In the model, a few cerebral columns have activity at a high level at each moment in time. The high-level activity is short-lasting and a column has activity at a comparatively low level almost all the time (Johannisson, 1984).

The two levels of activity are separate because between them, there is a gap in the activity range (Johannisson, 1993). In this context, activity at a high level in the columns is activity at a conscious level. When this activity is observed from outside, it is seen as neuronal activity, and when the same activity is observed from inside, it is seen as mental activity.

Each column has a set of connections with other columns and other parts of the nervous system. These connections constitute the functional properties that characterize every single column (Johannisson, 1984). The various connections are of different strengths, and association paths in cognitive processes usually follow the strongest connections. However, weak outgoing connections from two or more columns with high activity may converge on a new column and produce additive effects, such that they can sometimes be stronger together than a strong connection from a single column (Johannisson, 1984; Johannisson, 1993). The association path then takes another direction rather than simply following the most obvious connection.

A direct recording of high neuronal activity should be possible if a recording electrode is placed close to a cerebral column. However, high activity in a column is a rare event because there are very many columns. This direct approach is also problematic because a recording of high activity may be difficult to repeat. Moreover, if high activity is recorded, it has to be distinguished from other types of high activity, such as that caused by damage from the recording electrode.

Alpha waves are not direct recordings of high activity in the cerebral columns. On the contrary, alpha waves are thought to represent neuronal activity at the low level, albeit from many cerebral columns (Johannisson & Nilsson, 1996). Small but synchronous changes up and down in the neuronal activity within the low activity range produce the alpha waves in this model. Nonetheless, the alpha waves reflect the regulating signals that keep the number of highly active columns within proper limits.

The two levels hypothesis is falsifiable. In this paper, the focus is on the existence of specific alpha groups, because without them the hypothesis cannot be true. At the outset of the study, the existence of these groups seemed unlikely because the prevailing opinion is that the distribution of individual alpha frequencies is bell-shaped and without gaps (Klimesch, 1999; Robinson, 2006; Fink & Neubauer, 2008; Başar, 2012; Bazanova & Vernon, 2014). The present study had to include measurements from a relatively large number of individuals before the alpha groups could be clearly seen (Fig. 3A).

Testing

The alpha groups are accessible for rigorous testing. The initial finding of three distinct alpha groups was made in a pilot study of 213 recordings from an EEG clinic (T Johannisson, 1999, unpublished data). The result was confirmed in the present study with an entirely different sample.

The individual alpha frequency can be measured in different ways (Başar, 2012; Bazanova & Vernon, 2014). Any method can be used to test the existence of the alpha groups as long as it is sufficiently precise. Measurements from a power spectrum with very narrow bands ought to give results that are comparable to those based on measurements from long sequences of alpha waves, at least when the alpha waves are well developed.

A major problem with the power spectrum and other averaging procedures is that they include a large amount of uncertain data. For testing the alpha hypothesis, only sequences with regular alpha waves are relevant (Johannisson & Nilsson, 1996). Irregular activity and various artifacts should not be included.

Another and more serious problem with measurements based on averaging procedures is that they are affected by the amplitude. Long waves often have higher amplitude compared to short waves, and waves with high amplitude weight the average to a greater extent than do small waves. This causes a large systematic error in frequency measurements based on averaging procedures. Inaccuracy of this type is not a problem when using sequences of alpha waves, as in the present study, because frequency measurements from sequences of waves are not dependent on the amplitude of the waves.

A method based on the power spectrum of alpha activity is not more objective than a method using the longest wave sequences containing regular alpha waves. Both methods are easy to use when there are well developed alpha waves. Thus, in a power spectrum, there can be a clear peak, which simplifies the measurement of the alpha frequency. However, when the alpha waves are poorly developed, the power spectrum often contains more than one peak, and the peaks are usually broad and asymmetric. In such cases, it is not easy to determine how the measurement should be done.

Thorough testing requires that independent researchers repeat the experiment in order to confirm the results. When preparing for new studies, perhaps it can be of some help that all of the alpha sequences behind the histograms in Fig. 3 are reproduced in Data S1. This file shows how long the longest sequence is for every participant. The second and third longest sequences are often included and they give an idea of the size of the variation in the measurements, but only the longest sequence was used. Exactly how the alpha frequency was calculated is shown for every participant. Also, the amount of noise and the general quality of the recordings can be assessed from this file.

Four temperaments

A bimodal distribution of the alpha frequencies in the main group around 10 Hz was found in the present study (Figs. 3A–3B). The two subgroups were not predicted from the alpha hypothesis and they were not visible in the pilot study. Therefore, the finding of a bimodal distribution was not expected.

In the pilot study, most of the subjects were patients taking medication that might affect their alpha waves. There were no such complications in the present study, and this may explain why the shallow valley in the middle of the large group around 10 Hz can be seen in the present study but not in the pilot study. A re-examination of the recordings from the subgroups around 9.5 and 10.5 Hz confirmed the bimodal distribution in the present study.

The personality profiles characterized in Table 1 and Fig. 6 can be recognized as four types of temperament that resemble the four classical temperaments. Many scientists consider the classical temperaments to be obsolete, and though there are many descriptions of these temperaments, there are no useful definitions. Thus, it is not possible to test whether the four profiles found in the present study are identical to the four classical temperaments.

It has previously been suggested that the definitions of the four classical temperaments use a combination of two personality dimensions (Robinson, 2001). This method of defining the temperaments is very elegant, but it does not take into account that the main differences for extraversion and neuroticism were found when the centers of the groups were compared with the surrounding zones (Figs. 4 and 5A). For example, if the sanguine temperament is defined as having high extraversion and low neuroticism (Robinson, 2001), then people from the centers of all four groups would be included (Fig. 4). However, this idea is not altogether ineffective because extraversion was slightly higher in the group at 8 Hz compared to the other three groups, and neuroticism was slightly lower in the group at 8 Hz than it was in two of the other three groups (Fig. 5C).

As another example, consider the case in which the melancholic temperament is defined as having low extraversion and high neuroticism (Robinson, 2001). In the present study, extraversion was slightly lower in the group at 12 Hz compared to the other three groups, while neuroticism was slightly higher in the group at 12 Hz than it was in two of the other three groups (Fig. 5C). Thus, the definition of the melancholic temperament based on two personality dimensions has some support when the comparison is made among the four groups. The problem with this definition comes to light when people from the surrounding zones are included, because they usually have low scores on extraversion and high scores on neuroticism (Fig. 5A).

Instead of having a parallel terminology with new names (such as 8, 9.5, 10.5, and 12), it is proposed that the temperaments be defined by the alpha frequency. The sanguine temperament would then refer to the group around 8 Hz, the phlegmatic temperament to the group at 9.5 Hz, the choleric temperament to the group at 10.5 Hz, and the melancholic temperament to the group around 12 Hz.

The four temperaments refer to the centers of the alpha groups. Both the centers and the surrounding areas should be included when the alpha waves are correlated to personality traits. Thus, the best description of the link between alpha waves and personality traits is provided by the curved lines in Figs. 4 and 6.

An attempt is made in the remaining part of ‘Discussion’ to explain the nature of the underlying relationship between alpha waves and personality traits. The two levels hypothesis (Johannisson, 1984; Johannisson, 1993) and the subsequent alpha hypothesis (Johannisson & Nilsson, 1996) are applied fully, and their implications in various fields are discussed.

Number of highly active columns

When the alpha frequency is 10 Hz, it has been estimated that the number of highly active columns alternates between 4 and 5 (Johannisson & Nilsson, 1996). The large group at 8 Hz has alpha waves that appear when the number of columns is switching between 3 and 4 (Johannisson & Nilsson, 1996; Johannisson, 1997). In a similar way, the group at 12 Hz is formed by alpha waves that are generated when the number of columns at the high level is alternating between 5 and 6 (Johannisson & Nilsson, 1996; Johannisson, 1997).

In addition to the alpha groups, there are other frequency groups. Infants exhibit waves that have a frequency of around 4 Hz, which may appear when the number of highly active columns is alternating between 1 and 2 (Johannisson, 1997). The number may increase with age, and when it alternates between 2 and 3, the waves become shorter and form a group around 6 Hz. If the increase in number continues, the limits become 3 and 4, and the frequency group is at 8 Hz. Some individuals stay within this sanguine group, while for others, the limits are raised further to 4–5 or 5–6. When awake, very few individuals have the limits 6 and 7, which give a frequency of around 14 Hz.

During the different sleep stages, all of the frequency groups described above may appear (Johannisson, 1997). Moreover, a frequency around 1 Hz may appear when the limits are 0 and 1 during deep or very deep sleep. Thus, adults have regulating systems that retain the capacity to keep the number of highly active columns within several different limits.

Regulating system

When the activity in a column jumps up to the high level, the upper limit for the number of highly active columns is reached. The regulating system has to react quickly to decrease the excitability of all of the columns with low activity (Johannisson & Nilsson, 1996). If there is a delay, additional columns can change their activity from the low to the high level and the number will exceed the upper limit.

Occasionally, two new columns may change the activity from the low to the high level at the same time. When this occurs, the number will exceed the upper limit. At this point, the regulating system has to decrease the excitability of the remaining columns and wait until two columns have stopped exhibiting high-level activity. If the regulating system fails and increases the excitability already when the activity in only one column has left the high level, the system can become trapped in repeated attempts to adjust the number. Strong regulating signals that are repeated with very short intervals damage the thinking processes and may explain what happens during a grand mal seizure.

The regulating system has to be very sensitive and react extremely quickly, which may lead to mistakes. If a decrease in excitability starts before a new column has changed the activity from the low to the high level, the number can go below the lower limit. The implication from this reasoning might be that in a grand mal seizure, the number of highly active columns is alternating between the upper limit and one step higher, while in a petit mal seizure, the number is alternating between the lower limit and one step lower.

However, under normal conditions, the number is kept within proper limits. This number is the first of two basic parameters in the model, and the second is the duration of high-level activity of a column (Johannisson & Nilsson, 1996).

Different durations

The limits for the number of highly active columns are the same in the groups around 9.5 and 10.5 Hz. The difference between these groups lies in the duration of high-level activity. In other words, different durations may account for the underlying difference between people with a phlegmatic temperament and those with a choleric temperament.

The alpha hypothesis makes it possible to calculate the duration of high-level activity in a column (Johannisson & Nilsson, 1996). The duration is the mean number of highly active columns times the wavelength. As an example, the mean number is 4.5 when the number of highly active columns is alternating between 4 and 5. The wavelength for an alpha wave is not the usual spatial period, but a temporal period, and therefore the inverse of the alpha frequency. For example, if the frequency is 10.0 Hz, then 1 divided by 10.0 /s gives the wavelength 100 ms. From these data, the duration of the activity at the high level in a column can be calculated as 4.5 × 100 ms = 450 ms.

When the frequency is 9.5 Hz, the duration is approximately 4.5 × 105 ms ≈ 470 ms. In the same way, a frequency of 10.5 Hz gives the duration ≈4.5 × 95 ms ≈ 430 ms. Differences in the duration thus account for the spreading of the frequency within the alpha groups.

Some individuals have alpha frequencies outside the main groups, for example, around 9 or 11 Hz. Although the computation is more complicated for alpha waves that are not within the main groups, it is still possible to calculate both the number and the duration if there are two stable rhythms available (Johannisson & Nilsson, 1996). In EEGs, switching between two frequencies is not uncommon, especially in young adults. Sufficient information is also available in sleep EEGs because several rhythms appear during sleep.

It is important to be able to measure the duration of high-level activity in a column because this duration determines the speed of the thinking processes. A relatively short duration, as seen in the choleric group, designates a fast turnover of highly active columns and rapid thinking. The opposite is seen in the phlegmatic group, where slow but reliable thinking processes are the result of a long duration and a slow turnover.

Differences between cholerics and phlegmatics in the duration of the high activity of the columns can explain the different mean scores observed in the cautiousness subdomain (Table 1). Short duration and speedy thinking can lead to a high degree of impulsiveness.

In the anxiety subdomain, the phlegmatic group had a somewhat lower score than did the choleric group (Table 1 and Fig. 6). This may be an effect of the relatively long duration in the phlegmatic group.

Number and duration together

In this paper, an alpha hypothesis (Johannisson & Nilsson, 1996) is used to explain the link between alpha waves and personality traits. Starting from measurements of the alpha frequencies, the two basic parameters in the hypothesis were calculated and the result is shown in Fig. 7.

When the number of highly active columns alternates between 3 and 4, the duration of high-level activity can be relatively short (Fig. 7). Conversely, when the number alternates between 5 and 6, the duration is comparatively long. A high number may give a flow of thoughts that can be complicated. A prolonged duration makes it possible to handle complicated thoughts because this duration results in slow but reliable thinking processes.

Figure 7 Number–duration diagram.

The x-axis shows the mean number of columns with activity at the high conscious level and the y-axis shows the mean duration of high-level activity in a column. One data point per participant. The ellipses indicate four types of temperament, and, as seen in the figure, the number and duration are somewhat different for these four. The sanguine temperament (in red) is characterized, in relative terms, by a low number and a short duration. Both the phlegmatic temperament (in green) and the choleric temperament (in yellow) have a medium number, but they have different durations. Furthermore, the melancholic temperament (in blue) can be seen as the result of having a high number and a long duration.

Figure 8 Summary illustrating how alpha waves, temperament types, and personality dimensions are interrelated.

The four temperaments are associated with four combinations of number and duration. These combinations are shown as ellipses in Fig. 7. Outside the ellipses, in all directions, the values for number and duration are suboptimal. Anxiety, depressed mood, and other symptoms may appear in the surrounding zones (Fig. 6). Thus, the number–duration diagram opens up for a new systematic way of understanding different mental disorders.

Conclusions

1. Alpha waves from different individuals can be categorized into three main groups where the frequencies are around 8, 10, and 12 Hz. The middle group appears to have a bimodal distribution, such that subdivision produces a total of four alpha groups.

2. When alpha waves are correlated with personality dimensions, the alpha frequency range is better divided into three segments. In the centers of the main alpha groups, extraversion is high and neuroticism is low.

3. There are many small differences in personality traits among the four groups. When taken together, the groups comprise four personality profiles that are similar to the four classical temperaments.

4. The link between alpha waves and personality traits (Fig. 8) implies an underlying relationship. There are two hypotheses that can be applied. One of these deals with the general organization of the forebrain and the other explains why alpha waves are generated.

Supplemental Information

Data S1 Supplemental EEG file

Click here for additional data file.

Data S2 Supplemental personality file

Click here for additional data file.

I would like to thank Jonatan Wistrand for the valuable discussions.

Additional Information and Declarations

Competing Interests

Author Contributions

Human Ethics

Data Availability

The author declares there are no competing interests.

Tomas Johannisson conceived and designed the experiments, performed the experiments, analyzed the data, contributed reagents/materials/analysis tools, wrote the paper, prepared figures and/or tables, reviewed drafts of the paper.

The following information was supplied relating to ethical approvals (i.e., approving body and any reference numbers):

Approval was obtained from the Regional Ethics Board for University of Göteborg (R627–97) and written informed consent was obtained from all participants. For participants younger than 18 years, written informed assent was also obtained from a parent.

The following information was supplied regarding data availability:

The raw data has been supplied as a Supplemental Dataset.

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
