# Peer review of "Correlations between personality traits and specific groups of alpha waves in the human EEG"

_PeerJ, doi:10.7717/peerj.2245_

## Round 0.1 · original submission · Major Revisions

Please revise according to the comments of the reviewers.

Reviewer 1 ·

Basic reporting

Comments on "Figures should be relevant to the content of the article, of sufficient resolution, and appropriately described and labeled":
1. Figure 3, please indicate bin width.
2. In Figure 6, in addition to the fitting curve, the author should also provide the data that used for fitting in the figure.


Comments on "All appropriate raw data has been made available":
3. The author mentioned a pilot study of 213 recordings from an EEG clinic. These data are neither provided nor published. The author should provide the raw data or state that it is unpublished data.

Experimental design

Comments on "Methods should be described with sufficient information to be reproducible by another investigator":
4. In page 6, line 154, the author should provide detailed information on the procedure to perform age and gender adjustment in order for other scientists to repeat the experiment.
5. In page 11, line 280, the author should be more specific about how the three curves in Fig 6A were considered together. Is there a combined score calculated and if so, how?
6. The definition for center and surrounding area in Figure 4 and Figure 5 seems arbitrary. Please provide a rational for how the range was defined to be center or surrounding, especially when there are gaps between center and surrounding (are those data in the gap discarded?).

Validity of the findings

Comments on "The data should be controlled":
7. A basic assumption of the current study is that individuals have distinct Alpha Frequency that does not change over time or with other external/internal factor. However, there is debate on this idea in the field. Therefore, it would be nice if the author could repeat the measurement on the same individuals at different time (e.g., after three months) to show that the peak of the same individual does not change at different measurements. Or, if similar study has been done to show that individual alpha frequency does not change at different measurements, the author should cite the reference.
8. The data analysis should be controlled for age, gender, and occupation of participants. For example, it is shown that alpha frequency gets lower when individual gets older (Clinical Neurophysiology 117 (2006) 1518–1528.). Therefore, I would appreciate it if the author could analyze the age profile in each alpha group and compare it with the age profile of all participants to make sure there are no differences. This will confirm that there is no enrichment of a certain age in a certain alpha group. In other words, it is important to rule out other possible factors that contribute to the separation of three (or four) alpha groups before attribute it to the four personalities.

Comments on "The data should be robust, statistically sound":
9. In Figure 4, it is smart to fit the mean values, which reduces the number of data points and makes the trend clearer. However, considering the wide distribution of data at each frequency, it would make the data representation more meaningful if the author could show the error bar of each mean value on Figure 4.
10. In Figure 5, please report sample size of each column. I would also appreciate it if the author could provide error bar on each column. Accurate reporting of the statistics of the data will make the conclusion from the data more meaningful.
11. In Table 1, please report standard deviation for each data point.

Comments on "Speculation is welcomed, but should be identified as such".
12. Figure 6B does not show a bimodal distribution that supports further division into sub-groups and thus the conclusion that the four frequencies represent four personalities. As mentioned in comment 8, without controlling for other factors, it is dangerous to make the conclusion that the four temperaments correlate with four alpha frequencies. The author also mentioned in discussion at line 381 that it is impossible to test them. Therefore, it should be made clear that this is just a speculation at this stage of research.

Additional comments

By measuring single sequences of alpha waves of 200 individuals, the author showed that alpha frequency can be divided into three main groups and four sub-groups. Comparison of alpha frequency to personality measurements revealed a linked between alpha waves and personality traits. The conclusion is supported by the data presented and the results provide new insight into the understanding of the relationship between alpha wave and personality dimensions. Therefore, I recommend publication of this manuscript. However, minor revision is suggested (please see comments for each section above) especially on the part of the statistical representation of the data, which is not sufficient in the manuscript.

Reviewer 2 ·

Basic reporting

The article was written in good English and well-organized. One minor point regarding data S2, it is a bit of unclear about the data structure.

Experimental design

The author provided detailed information about experimental design, including subjects, data collection and analysis. A minor point about subjects, what's the meaning of including that wide range of age differences (15-70)? Or please provide the number of different age group, like teenagers(below 18), young adult (19-30)...

Validity of the findings

The author presented plenty of data to visualize the correlation between alpha frequency and personality traits, which brought new insights of this relationship and also discussed two potential hypothesis. For the part of discussion, it is a bit of confusing about the what is the main point that author want to discuss, the cor-relationship or two level hypothesis or another hypothesis. Author should consider to shorten some unimportant parts and focus on major discussion.

Reviewer 3 ·

Basic reporting

This is a well writing manuscript with some interesting results. It give us a obvious correlation between personality trait and

EEG wavelength. Some minor revise should be done before go to next steps.
1. How to invited these participants to join this research work? By advertisement or any other way? Where did these

participant come from? from the university or any other place?

2. in line 133, "...conductive gel was used and very low impedance values were usually obtained.",Please exhibite the

impedence values which was used, such as <5kQ or <10 kQ, and so on.

3. in line 157-158, "... on average, higher extraversion scores and lower neuroticism scores. In addition, some of the mean

estimates for the subdomains were farther from 50% than others." From this sentence, it can be imagined that tese

participants have some special personality charateristic comparing to normal population? And that means this sample is not a

normal sample in general speaking? Why?
4. How to analysis the EEG data, and which equipment was used to collected EEG data, Please give some information about

these.
5. in figure 3. Please note the total number of participants in figure A,B and C.
6. in figure 7. Please give the names of the four types of temperament.
7. Please add on a table to discribed the demographic information of these participants.

Experimental design

This research have a good design.

Validity of the findings

The data on which the conclusions are based be provided available in an acceptable discipline-specific repository.

Additional comments

1. How to invited these participants to join this research work? By advertisement or any other way? Where did these

participant come from? from the university or any other place?

2. in line 133, "...conductive gel was used and very low impedance values were usually obtained.",Please exhibite the

impedence values which was used, such as <5kQ or <10 kQ, and so on.

3. in line 157-158, "... on average, higher extraversion scores and lower neuroticism scores. In addition, some of the mean

estimates for the subdomains were farther from 50% than others." From this sentence, it can be imagined that tese

participants have some special personality charateristic comparing to normal population? And that means this sample is not a

normal sample in general speaking? Why?
4. How to analysis the EEG data, and which equipment was used to collected EEG data, Please give some information about

these.
5. in figure 3. Please note the total number of participants in figure A,B and C.
6. in figure 7. Please give the names of the four types of temperament.
7. Please add on a table to discribed the demographic information of these participants.

---

## Round 0.2 · accepted · Accept

Congratulations on this work.

Reviewer 2 ·

Basic reporting

No Comments

Experimental design

No Comments

Validity of the findings

No Comments